Cyy-287, a novel pyrimidine-2,4-diamine derivative, efficiently mitigates inflammatory responses, fibrosis, and lipid synthesis in obesity-induced cardiac and hepatic dysfunction

Ni Jinhuan 1
Zhang Xiaodan 1
Huang Huijing 1
Ni Zefeng 2
Luo Jianchao 1
Zhong Yunshan 1
Hui Min 1
Liu Zhiguo 2
Qian Jianchang 1 qianjc@wmu.edu.cn
Zhang Qianwen 1 zqwlovelove@126.com
1 Institute of Molecular Toxicology and Pharmacology, Wenzhou Medical University , Wenzhou , China
2 Chemical Biology Research Center at School of Pharmaceutical Sciences, Wenzhou Medical University , Wenzhou , China
Uversky Vladimir
Electronic publication date: 2024 Feb 29
Publication date: 2024
Volume: 12
Electronic Location ID: e17009
Received 2023 Oct 17; Accepted 2024 Feb 5
Copyright: © 2024 Ni et al.
Copyright year: 2024
Copyright holder: Ni et al.
License: This is an open access article distributed under the terms of the Creative Commons Attribution License, which permits unrestricted use, distribution, reproduction and adaptation in any medium and for any purpose provided that it is properly attributed. For attribution, the original author(s), title, publication source (PeerJ) and either DOI or URL of the article must be cited.
License URL: https://creativecommons.org/licenses/by/4.0/

Keywords: Cyy-287, Inflammation, Obesity complications, Ventricular remodeling, Hepatic steatosis, NF-kB, AMPK

Funding: Wenzhou Municipal Science and Technology Bureau 2020Y1214 Natural Science Foundation of Zhejiang Province LQ21H310007 National Natural Science Foundation 82203839 and 81973168 This work was supported by the grants from the Wenzhou Municipal Science and Technology Bureau (No. 2020Y1214), the Natural Science Foundation of Zhejiang Province (No. LQ21H310007), and the National Natural Science Foundation of China (Nos. 82203839 and 81973168). The funders had no role in study design, data collection and analysis, decision to publish, or preparation of the manuscript.

==============================
Background

Inflammation and metabolic disorders are important factors in the occurrence and development of obesity complications. In this study, we investigated the protective effect and underlying mechanism of a novel pyrimidine-2,4-diamine derivative, Cyy-287, on mice fed a high-fat diet (HFD).

Methods

The mice were randomly separated into four groups (n ≥ 7): control (regular diet), HFD, HFD with Cyy-287 (5 mg/kg), and HFD with Cyy-287 (20 mg/kg) following HFD feeding for 10 weeks. After a 10-week administration, ALT and AST enzymes, echocardiography, immunohistochemical (IHC), Western blot (WB), Masson and Sirius Red staining were used to evaluate functional and morphological changes to the heart and liver. Microsomes from the mouse liver were extracted to quantify the total amount of CYP450 enzymes after drug treatment.

Results

Cyy-287 decreased the levels of serum glucose, LDL, TC, ALT, and AST activities in HFD-treated mice. However, Cyy-287 administration increased ejection fraction (EF) and fractional shortening (FS) index of the heart. Cyy-287 inhibited histopathological changes in the heart and liver; decreased inflammatory activity; significantly diminished p38 mitogen-activated protein kinase (MAPK), the nuclear factor-kappa B (NF-κB) axis, and sterol regulatory element-binding protein-1c (SREBP-1c); and upregulated the AMP-activated protein kinase (AMPK) pathway in HFD-treated mice. Cyy-287 restored the content of hepatic CYP450 enzymes.

Conclusion

These findings demonstrated that Cyy-287 protected heart and liver cells from obesity-induced damage by inhibiting inflammation, fibrosis, and lipid synthesis.

Introduction

Obesity is a widespread public health problem that is often accompanied by a variety of complications, such as cardiovascular and cerebrovascular diseases, nephropathy, diabetes, and fatty liver (Caballero, 2019; Polyzos, Kountouras & Mantzoros, 2019; Piche, Tchernof & Despres, 2020). All of these complications pose a serious threat to the health of obese patients and reduce their quality of life. Although weight control can alleviate its progression, drug treatment is still one of the most efficient ways to alleviate the development of obesity complications (Jackson et al., 2015; Narayanaswami & Dwoskin, 2017; Srivastava & Apovian, 2017). Current basic treatment strategies include blood pressure reduction, lipid reduction, and hypoglycemic treatment (Tanaka, 2020; Klop, Elte & Cabezas, 2013). However, these treatments cannot block the progression of the disease completely. Therefore, it is necessary to determine new therapeutic strategies and develop new therapeutic drugs.

Chronic inflammation, intracellular mitochondrial dysfunction, insulin resistance, metabolic disorders, autophagy, and endoplasmic reticulum stress are important mechanisms that induce obesity complications (Ye, 2021; Fu et al., 2011; Goldman, Zhang & Jin, 2010; Aron-Wisnewsky et al., 2020). Among these, the inflammatory response is a crucial process in the development of tandem pathologies (Cox, West & Cripps, 2015). Nuclear factor-κB (NF-κB) regulates a large number of genes involved in different immunomodulatory responses. The mechanism of NF-κB activation is the inducible degradation of IκBα triggered through its site-specific phosphorylation by a multi-subunit IκB kinase (IKK) complex. IKK can be activated by various factors, including cytokines, growth factors, mitogens, and stress agents (Nguyen & Stamper, 2017). Recently, it has been suggested that sporoderm-broken spores of Ganoderma lucidum (BSGLP) can inhibit the upregulation of the NF-κB signaling pathway in adipose tissue induced by a high-fat diet (HFD), thus suppressing obesity and hyperlipidemia by modulating inflammation, gut microbiota, and intestinal tract barrier function (Sang et al., 2021). Another study has confirmed that controlling adipose inflammation mediated by adipose tissue macrophages can improve metabolic dysfunction and reduce obesity (Wu et al., 2021). Consequently, blocking the inflammatory response by targeting the NF-κB pathway may be an effective treatment against obesity complications.

AMP-activated protein kinase (AMPK), a serine/threonine protein kinase, senses cellular energy status and is activated by higher AMP/ATP or ADP/ATP ratios (Lv et al., 2019). Its functions extend to various pathways, including those linked to metabolic diseases (Fogarty et al., 2016). Emerging evidence shows that AMPK activation is beneficial for the treatment of several metabolic diseases, including obesity and nonalcoholic fatty liver disease (NAFLD) (Garcia et al., 2019). Additionally, several natural compounds and synthetic drugs have been recognized to prevent obesity-related renal injury, cardiomyopathy, and hepatic damage by activating AMPK and its downstream pathways (Li et al., 2021, 2022). AMPK activation can effectively prevent inflammation-induced hepatic steatosis though the IKK/NF-κB signaling pathway (Li et al., 2018; Jiang et al., 2021). Based on these studies, we speculate that targeting the AMPK pathway could represent a new potential strategy for obesity complications.

Previously, we investigated the antitumor effect and underlying mechanism of a novel pyrimidine-2,4-diamine derivative, Cyy-287 (Fig. 1A), in non-small cell lung cancer (Zhang et al., 2022). Due to the fact that C57BL/6J mice exhibited comparable phenotypes to those found in human diabetes and metabolic disorders following a HFD, we developed a mouse model of high-fat diet-induced obesity using C57BL/6J mice. We found that the differential genes between the control group and the Cyy-287 group were enriched in the inflammatory pathway. In this study, our data suggested that Cyy-287 could efficiently block the chronic inflammation and de novo lipogenesis in HFD-treated mice, and its mechanism might be related to the MAPK/NF-κB and AMPK/SREBP-1c pathways.

Figure 1 Cyy-287 restrains the occur of hyperglycosemia and hyperlipidemia induced by HFD in mice.

(A) The chemical structure of cyy-287. (B) Body weight changes in mice. (C) The BMI of mice. The serum glucose level (D) and lipid markers LDL (E) and TC (F) of mice treated under different conditions was measured. Data are presented as the mean ± SEM (n = 3). ***P < 0.001, ****P < 0.0001 compared with the control group. #P < 0.05, ##P < 0.01, ###P < 0.001, ####P < 0.0001 compared with the HFD group.

Materials and Methods

Chemicals and reagents

Cyy-287 was synthesized as previously described by our laboratory with a purity of over 98% (Zhang et al., 2022). Antibodies against p38 (#8690), P-p65 (#3033), P-AMPK (#2535), P-p38 (#4511)), and P-JNK (#4668) were purchased from Cell Signaling Technology (Danvers, MA, USA). Antibodies against Collagen-I (#14695-1-AP) and tubulin (#11224-1-AP) were purchased from Proteintech (Wuhan, China). Anti-SREBP-1c (#abs152294) was obtained from Absin (Shanghai, China). Antibodies for immunohistochemical (IHC) staining including anti-F4/80 (#ab300421) and anti-TNF-α (#ab183218) were obtained from Abcam (Cambridge, UK). The secondary antibody (#111-585-003) for Western blot (WB) was acquired from Jackson ImmunoResearch (West Grove, PA, USA) and for IHC (#GB23301; #GB23303) from Servicebio (Wuhan, China). Hematoxylin (#G1005), Masson (#G1006), and Sirius Red (#G1018) dye solution used for staining of tissue sections were purchased from Servicebio (Wuhan, China).

Animal models

All animal care and experimental procedures were approved by the Animal Policy and Welfare Committee (IACUC) of Wenzhou Medical University (approval ID: WYDW-2019-0809) and conformed to the provisions of the Declaration of Helsinki. A total of 36 male C57BL/6J mice weighing 18–22 g were purchased from Beijing Vital River Laboratory Animal Technology Co., Ltd. (Beijing, China) and were all housed in an environmentally controlled room at a temperature of 20 ± 2 °C, humidity of 50 ± 5%, and light/dark cycle of 12 h. The mice had unrestricted access to food and water. After adaptive feeding for 7 days, the mice were randomly divided into four groups (n ≥ 7): (1) the control group (n = 7); (2) the HFD group (n = 9); (3) the HFD group + Cyy-287 (5 mg/kg) (n = 10); and (4) the HFD group + Cyy-287 (20 mg/kg) (n = 10). The entire HFD group was fed with a HFD (#D12492) from Research Diets, Inc. (New Brunswick, NJ, USA), consisting of 60% fat, 20% protein, and 20% carbohydrates, while the control group was fed with a standard diet (#MD17122) from Medicience Ltd. (Yangzhou, China), consisting of 10% fat, 20% protein, and 70% carbohydrates. After feeding respectively for 10 weeks, Cyy-287 was dosed orally once daily for another 10 weeks, and mice in the control group were given equal volume of normal saline by oral administration. Body weight was recorded every week. On the 20th week, the blood glucose of mice was recorded with a blood glucose meter (#GA-3) from Sinocare Inc. (Changsha, China), and echocardiography was performed. Finally, serum samples and liver and heart tissues were collected after the mice were euthanized through gradual CO2 asphyxiation and cervical dislocation.

Measurement of biochemical parameters

The liver tissues were homogenized in physiological saline (1:9 w/v) under ice-water bath conditions. After centrifugation for 10 min, we used the supernatant diluted 50 times to test the concentration of ALT, AST with ALT/GPT (#C009-2-1), and AST/GOT (#C010-2-1) kits (Jiancheng Bioengineering Institute, Nanjing, China). After the blood samples with heparin as anticoagulant were centrifuged at 3,000 rpm at 4 °C for 10 min, supernatant was used to detect serum lipids. The TC and TG detection was performed by Servicebio (Wuhan, China) using Chemray 800 (Redu Life Sciences Co., Ltd., Shenzhen, China).

Echocardiography

After mice were anesthetized with isoflurane, the ultrasound probe was positioned on the left chest to capture the parasternal long-axis image, which included the mitral valve and aortic valve. A short-axis image of the left ventricle was then obtained perpendicular to the long axis. The Vevo-3100 high-resolution imaging system (Fujifilm VisualSonics, Bothell, WA, USA) was used to evaluate parameters of the heart’s diastolic function, such as ejection fraction (EF) and fractional shortening (FS).

Western blot analysis

Mouse tissues were lysed in RIPA lysis buffer (#P0013C, Beyotime Biotechnology, Shanghai, China). Protein contents were measured using BCA assay (#P0012S, Beyotime Biotechnology, Shanghai, China). The protein samples were then subjected to 10% SDS-PAGE and transferred to a 0.45 μM nitrocellulose filter membranes (PALL, Port Washington, NY, USA). After blocking with 5% skim milk in Tris buffered saline containing 0.05% Tween 20 (TBST, pH 7.4), for 1.5 h at room temperature, the membranes were incubated with antibodies (1:1,000) at 4 °C overnight. Membranes were then incubated with the corresponding HRP- conjugated secondary antibody (1:10,000; Jackson Immunoresearch, West Grove, PA, USA) for 2 h. The protein bands on membranes were detected using an enhanced chemiluminescence detection kit (Millipore, Burlington, MA, USA) and the protein gray value was quantified using ImageJ software (https://imagej.nih.gov/ij/).

Histological and immunohistochemical analyses

Tissue samples were fixed with 10% formalin, embedded in paraffin, and sectioned to an 8 μm thickness slide. Hematoxylin and eosin (HE) stain and Masson’s trichrome stain were then used to assess the extent of tissue fibrosis. In addition, heart slides were stained with Sirius Red and liver slides were stained with Oil Red to appraise hepatic steatosis. Images were captured using a Nikon Eclipse E100 ortho optical microscope (Nikon, Japan). For IHC, paraffin sections were dewaxed and rehydrated. Antigens were retrieved with citric acid antigen repair buffer and then put into 3% hydrogen peroxide solution to eliminate endogenous peroxidase. The slides were then blocked with 3% BSA and incubated with anti-TNF-α or F4/80 antibody (1:100) at 4 °C overnight. After incubating with HRP-labeled secondary antibody (1:1,000, Servicebio, Wuhan, China), the samples were detected using DAB staining. Images were acquired and analyzed using a Pannoramic 250 FLASH scanner (3D HISTECH, Budapest, Hungary).

Extraction of live microsomes and determination of P450 enzyme content

The liver tissues were homogenized in 2.5 mL per gram PBS-Sucrose (0.25 M) buffer. The supernatant was harvested after centrifugation at 11,000 rpm for 15 min at 4 °C twice. The supernatant was centrifuged at 75,600 ×g in ultracentrifuge for 2 h at 10 °C. The precipitate was resuspended with PBS by pipetting until the solution was clear. We then determined the cytochrome P450 enzyme (CYP450) content in the microsomal samples using the CO differential method. We took 1 mL of microsome samples which were diluted to 1 mg/mL with PBS, aerated with carbon monoxide for 2 min, and then added 5 mg of sodium hydrosulfite. The absorbance at 450 and 490 nm was measured by Evolution 201 ultraviolet-visible spectrophotometer (Thermo Fisher Scientific, Waltham, MA, USA). The calculation formula was: C (nmol/mL) = (OD 450 nm–OD 490 nm) ⁎ 1,000/91.

Statistical analysis

All data are presented as mean ± standard error of the mean (S.E.M). We first conducted one-way analysis of variance (ANOVA) to determine whether there was a significant effect of the independent variable on the response variable. Subsequently, we conducted Tukey’s multiple comparisons tests to generate adjusted P-values for different pairwise comparisons. If the variance was uneven, we used Welch’s ANOVA and Tamhane’s T2 test for multiple comparisons. P < 0.05 was considered statistically significant (GraphPad Prism 6.0 and SPSS 22.0 software).

Results

The improvement of Cyy-287 on obesity-induced metabolic disorder

To confirm whether obesity causes multiple organ damage, particularly cardiac and hepatic dysfunction, and to evaluate the preventive effect of Cyy-287 treatment, we established a HFD-treated mice model. After 20 weeks, mice fed with HFD showed a higher rate of weight gain (>20 g) than mice fed with standard diet (<5 g), and each of their weights exceeded 45 g (Fig. 1B). The body mass index (BMI) of HFD-treated mice was significantly higher compared to the control group (P < 0.0001), but Cyy-287 treatment did not decrease BMI (Fig. 1C). In addition, the serum glucose level of mice in the HFD group increased significantly (P < 0.001), indicating the occurrence of hyperglycemia, and Cyy-287 significantly reduced blood glucose compared with the HFD group (5 or 20 mg/kg, P < 0.05; Fig. 1D). Additionally, Cyy-287 decreased serum lipid markers (LDL and TC) (LDL, 20 mg/kg, P < 0.001; TC, 20 mg/kg, P < 0.0001) in HFD-treated mice (Figs. 1E and 1F). These results showed that Cyy-287 improved metabolic disorders caused by obesity, including decreased serum glucose and decreased serum lipid production.

The preventive effect of Cyy-287 on obesity-linked cardiac and hepatic dysfunction

The mice in the HFD group showed obvious ventricular remodeling characteristics, such as increased ventricular inner diameter and different EF (P < 0.0001) and FS (P < 0.001), confirming cardiac dysfunction caused by obesity (Figs. 2A–2C). Cyy-287 significantly restored mice cardiac function indicators EF (20 mg/kg, P < 0.001) and FS (20 mg/kg, P < 0.01), and reduced cardiac hypertrophy index in a dose-dependent manner (Table 1). The levels of ALT and AST were also significantly increased in the HFD group compared with the control group (P < 0.0001), confirming the hepatic dysfunction caused by obesity (Figs. 2D and 2E). Cyy-287 exhibited a remarkably suppressive effect on the elevation of serum ALT (20 mg/kg, P < 0.001) and AST (5 or 20 mg/kg, P < 0.0001). These outcomes corroborate Cyy-287’s efficacy in impeding obesity-related deterioration of cardiac and hepatic functions.

Figure 2 Cyy-287 improves cardiac and hepatic dysfunction induced by HFD in mice.

(A) Representative echocardiography M-mode images in mice. (B and C) Quantification of cardiac function indexes. (D and E) Determination of AST and ALT. Data are presented as the mean ± SEM (n = 5). ***P < 0.001, ****P < 0.0001 compared with the control group. ##P < 0.01, ###P < 0.001, ####P < 0.0001 compared with the HFD group.

Table 1 Echocardiographic parameters of rats (n ≥ 5).

Parameter	Units	Control	HFD	Cyy-287 (5 mg/kg)	Cyy-287 (20 mg/kg)	
Diameter;d (LV Trace)	mm	3.7 ± 0.48	4.55 ± 0.26*	4.4 ± 0.36	4.16 ± 0.44	
Diameter;s (LV Trace)	mm	1.94 ± 0.69	3.44 ± 0.31***	3.2 ± 0.31	2.36 ± 0.48#	
EF (LV Trace)	%	78.98 ± 12.38	48.31 ± 6.56****	53.12 ± 5.58	74.74 ± 7.15###	
FS (LV Trace)	%	48.51 ± 12.2	24.34 ± 3.8***	27.26 ± 3.55	43.77 ± 6.85##	
V;d (LV Trace)	uL	59.58 ± 19.02	95.2 ± 12.77*	88.58 ± 16.89	77.86 ± 18.31	
V;s (LV Trace)	uL	14.03 ± 12.92	49.48 ± 11.01***	41.58 ± 9.66	20.53 ± 8.55##	
Notes:

EF, Ejection fraction; FS, Fractional shortening; V;d, Left ventricular diastolic volume; V;s, Left ventricular systolic volume.

* P < 0.05.

*** P < 0.001 compared with the control group.

**** P < 0.0001.

# P < 0.05.

## P < 0.01.

### P < 0.001 compared with the HFD group.

Cyy-287 inhibits obesity-induced cardiac inflammation by suppressing MAPK/NF-κB and activating AMPK pathways

To further investigate the effect of Cyy-287 on obesity-induced cardiac dysfunction, we conducted HE and IHC staining. The analysis of the histopathological images showed that HFD caused cardiac morphological changes, including myocardial fiber atrophy and disorder and inflammatory cell infiltration, which improved after Cyy-287 treatment (Fig. 3A). In addition, Cyy-287 treatment reduced the expression of TNF-α in HFD-treated mice. We then evaluated the degree of myocardial fibrosis by Masson and Sirius Red staining in mice. As shown in Figs. 3A–3C there was significant fibrous proliferation (Masson, P < 0.001) and collagen deposition (Sirius Red, P < 0.05) in the myocardium of HFD-treated mice, and the progression of myocardial fibrosis was significantly blocked after Cyy-287 treatment (Masson, 20 mg/kg, P < 0.01; Sirius Red, 20 mg/kg, P < 0.05). Consistent with the IHC results, Western blot results showed that Cyy-287 improved the HFD-induced Collagen-I expression (20 mg/kg, P < 0.05) (Fig. 3D). The HFD group demonstrated markedly elevated levels of phosphorylated p65 (RelA), a crucial subunit of NF-κB, and p38, a member of the MAPK family. Conversely, Cyy-287 treatment significantly attenuated the cardiac expression of phosphorylated p65 and p38 (20 mg/kg, P < 0.05), as compared to the HFD-only group. Meanwhile, the phosphorylation of AMPK was greatly inhibited by HFD (P < 0.05), and the cardiac expression of phosphorylated AMPK was effectively increased after Cyy-287 treatment (P = 0.30). Hence, our findings imply that Cyy-287 proficiently safeguarded the myocardium from harm via the modulation of MAPK/NF-κB and AMPK signaling cascades.

Figure 3 Cyy-287 attenuates myocardial inflammation and fibrosis by inhibiting p38 MAPK and activating AMPK in HFD mice.

(A) Tissue sections were analyzed by HE, Masson, and Sirus Red and IHC using anti-TNF-α antibody (scale bar: 50 μm). (B and C) Quantitative data of Masson and Sirus Red staining in (A). (D) Proteins were extracted from myocardium and then subjected to Western blot analysis. Quantified results were plotted. Data are presented as the mean ± SEM (n = 3). *P < 0.05, ***P < 0.001 compared with the control group. #P < 0.05, ##P < 0.01 compared with the HFD group.

Cyy-287 relieves obesity-induced liver damage and MAPK/NF-κB pathway protein expression and AMPK activity inhibition

Compared with the control group, the liver tissue weight of the HFD group was significantly increased (P < 0.01) and reversed after Cyy-287 treatment (20 mg/kg, P < 0.05; Fig. 4A). Examination of the histopathological illustrations unveiled the morphological alterations brought about by HFD in the liver, including hepatic steatosis, inflammatory responses within the hepatic lobules, centrilobular hepatocyte necrosis, and the presence of ballooned hepatocytes (Fig. 4B). Cyy-287 notably alleviated liver damage and reduced liver cell necrosis and degeneration. In addition, Masson and F4/80 staining showed that Cyy-287 could reduce hepatic fibrosis and macrophage infiltration compared to the HFD group (Masson, 20 mg/kg, P < 0.01; F4/80, 20 mg/kg, P < 0.001; Figs. 4C and 4D). Similar to the results in the heart, the phosphorylation of AMPK was greatly inhibited by HFD (P < 0.05), and the hepatic expression of phosphorylated AMPK was effectively increased after Cyy-287 treatment (20 mg/kg, P < 0.05; Fig. 4E). In addition, the hepatic expression of phosphorylated p65, p38, and JNK was effectively decreased after Cyy-287 treatment (20 mg/kg) compared to the HFD-only group (P < 0.05; P < 0.05; P < 0.001). In conclusion, our histological and Western blot analyses substantiate that orally administering Cyy-287 thwarted liver impairment and inflammation triggered by HFD.

Figure 4 Cyy-287 mitigates hepatic fat accumulation and damage through MAPK/NF-κB dowregulation and activating AMPK in HFD mice.

(A) The gross images of livers, and the weight of liver tissue was measured and plotted. (B) Tissue sections were analyzed by HE, Oil Red, and Masson and IHC using F4/80 anti-body (scale bar: 50 μm). (C and D) Quantitative data of Masson and F4/80 in (B). (E) Proteins were extracted from myocardium and then subjected to Western blot analysis. Quantified results were plotted. Data are presented as the mean ± SEM (n = 3). *P < 0.05, **P < 0.01, ****P < 0.0001 compared with the control group. #P < 0.05, ##P < 0.01, ###P < 0.001 compared with the HFD group.

Cyy-287 relieves obesity-induced liver lipid storage and SREBP-1c expression

SREBP-1c is a key regulatory transcription factor involved in de novo lipogenesis, and the over-expression of SREBP-1c was correlated with lipid storage in the liver. Oil Red staining showed that Cyy-287 could reduce fatty accumulation in liver induced by HFD (Fig. 4B). Furthermore, the HFD group exhibited a significantly elevated mRNA (P < 0.001; Fig. S1) and protein (P < 0.05; Fig. 4E) expression level of SREBP-1c compared to the control group. However, oral administration of Cyy-287 (20 mg/kg) effectively curbed the hepatic mRNA (P < 0.01) and protein (P < 0.05) expression of SREBP-1c in comparison to the HFD-only group. Our data categorically established that the hepatoprotective influence of Cyy-287 is due to its lipid synthesis inhibitory attributes.

Cyy-287 restores the content of hepatic CYP450 enzymes

As a type of haemoglobin, cytochrome P450 enzymes (CYP450) participate in the metabolism of many substances, including endogenous substances, exogenous substances, and drugs. CYP450 exists mainly in liver microsomes and the activity of CYP450 can be used to evaluate the liver function. Hence, we extracted mouse liver microsomes and quantified the total number of CYP450 enzymes. As shown in Fig. 5, the number of hepatic CYP450 enzymes in the HFD group was significantly reduced (P < 0.05), and the decrease of liver CYP450 enzymes induced by HFD was significantly reversed after Cyy-287 treatment (20 mg/kg; P < 0.05), which further indicated that Cyy-287 could restore liver function and alleviate liver injuries caused by high fat.

Figure 5 Cyy-287 increases the content of CYP450 enzymes in liver microsomes of HFD mice.

(A) Representative UV chromatograms of liver microsomes. (B) Quantification of liver microsomal content. Data are presented as the mean ± SEM (n = 3). *P < 0.05 compared with the control group. #P < 0.05 compared with the HFD group.

Discussion

Obesity has been on the rise worldwide over the past several decades and is now considered a public health epidemic. Studies have found that obese people present a chronic low-grade inflammatory state with an imbalance of pro-inflammatory and anti-inflammatory immune cells (Gregor & Hotamisligil, 2011). Obesity-induced inflammation leads to maladaptive responses such as fibrosis and necrosis that can cause significant tissue damage (Crewe, An & Scherer, 2017). Obesity-induced inflammation is unique in that it involves multiple organs, including adipose tissue, pancreas, liver, skeletal muscle, heart, and brain. Persistent metabolic inflammation also leads to serious syndromes such as insulin resistance, type 2 diabetes, cardiovascular disease, liver disease, cancer, and neurodegeneration (Saltiel & Olefsky, 2017). It has been reported that anti-inflammatory drugs or natural compounds can alleviate obesity-related metabolic diseases (Lee et al., 2019; Ngamsamer, Sirivarasai & Sutjarit, 2022). Consequently, blocking obesity-induced inflammation could be an effective way to prevent tissue damage and syndromes.

Here, we demonstrated that Cyy-287, a novel pyrimidine-2,4-diamine derivative, had a strong inhibitory effect on HFD-induced inflammation and tissue injury. The NF-κB pathway is a critical signaling axis that mediates the expression of inflammation-related mediators such as via NF-κB-binding motifs in their promoters (Dorrington & Fraser, 2019). NF-κB protein is mainly composed of two subunits, p50 and p65, which could be activated by cytokines, pathogens, and radiation (Liu, Zhang & Joo, 2017). In this study, we found that Cyy-287 can significantly inhibit NF-κB p65-mediated inflammatory pathways in the heart and liver tissues of obese mice. Oral administration of Cyy-287 to HFD-treated mice induced a noteworthy decline in the histopathological disruption of the heart and liver, characterized by attenuation in macrophage infiltration, fibrosis, and necrotic deterioration. Furthermore, Cyy-287 enhanced EF and FS and reduced serum AST and ALT levels due to liver functional impairment induced by HFD. Additionally, Cyy-287 effectively curbed abnormal hepatic lipid aggregation (TG). Our data supported the idea that Cyy-287 could effectively protect against obesity-induced cardiac and hepatic damage by inhibiting the NF-κB mediated inflammatory pathway.

As a fundamental transducer of upstream signaling for NF-κB, the MAPK family is intricately linked to cell death and is accountable for generating proinflammatory cytokines. Several studies have shown that MAPK is involved in oxidative stress and apoptosis, and blocking the MAPK signaling pathway can prevent paracetamol-induced hepatotoxicity by regulating pro-inflammatory cytokines (Noh et al., 2013). In addition, research has suggested that curcumin reduces chemical and drug-induced cardiotoxicity by inhibiting the p38 MAPK signaling pathway (Yarmohammadi, Hayes & Karimi, 2021). JNK is another member of the MAPK superfamily that affects insulin resistance and plays an important role in inflammatory response and oxidative stress. The persistent activation of JNK by inflammatory factors was observed in different tissues of obese subjects. Activated JNK acts on nuclear factor-κb (NF-κB) and activator protein-1 (AP-1) to produce more inflammatory factors, further reduce the sensitivity of insulin target cells to insulin, and eventually form a vicious cycle and aggravate insulin resistance (Bazin et al., 2012). Our Western blot data showed that HFD activated the expression of P-p38, leading to cardiomyocyte and hepatocyte injury. Cyy-287 effectively protected the heart and liver from damage by inhibiting the MAPK pathway.

As a highly conserved regulator of metabolism, AMPK is also a critical component in the prevention of various inflammatory signaling pathways, which may represent a larger therapeutic target for the control of insulin resistance, diabetes, and obesity (Noor et al., 2020). A recent study showed that the CaMKKβ/LKB1/AMPK axis and Ca2+ levels can provide a quick, adaptable switch to promote cell survival (Yang et al., 2020). In addition, activation of AMPK alleviates paracetamol-induced hepatotoxicity by inhibiting the release of pro-inflammatory cytokines and abnormal lipid metabolism (Jiang et al., 2021). In recent years, cardiomyopathy caused by obesity has attracted more and more attention. Studies have shown that upregulating the SIRT1/LKB1/AMPK pathway can reduce obesity-induced cardiomyopathy (Li et al., 2022). Our findings evinced that the elevation in the phosphorylation of AMPK was elicited by the administration of Cyy-287, as well as highlighted that the administration of Cyy-287 repressed obesity-related cardiac and hepatic injury by upregulating the AMPK signaling pathway.

It has been reported that the accumulation of liver fat will lead to inflammation and further liver damage. SREBP-1c is an important transcription factor in de novo lipogenesis and plays a key role in regulating liver fat metabolism. The entry of SREBP-1c into the nucleus is regulated by AMPK (Chao et al., 2019). Western blot results showed that Cyy-287 could regulate the AMPK-SREBP-1c signal pathway to improve metabolic disorders caused by HFD in liver. Oil Red staining results further confirmed that Cyy-287 alleviated the excessive accumulation of liver fat. In addition, the levels of lipid markers (LDL and TC) in HFD mice decreased after Cyy-287 treatment. Thus, our data suggested that Cyy-287 can effectively reduce liver lipid synthesis and fat accumulation in HFD mice by inhibiting SREBP-1c.

Unfortunately, our research still had some limitations. The specific mechanism of Cyy-287 changing AMPK activity has not been clarified, and the molecular mechanism has not been deeply studied. We will do further research in the follow-up study.

Conclusion

In this article, we provided evidence that Cyy-287 has a significant therapeutic effect on obesity-induced heart and liver damage by inhibiting inflammatory response and lipogenesis in HFD-treated mice. The mechanisms of action were revealed to involve Cyy-287’s potent anti-inflammatory properties, mediated by inhibiting the NF-κB and MAPK signaling pathways, activating AMPK signaling pathways, and suppressing liver lipid synthesis (Fig. 6). Therefore, Cyy-287 has the potential to be used as a therapeutic drug to reduce the complications of obesity.

Figure 6 The mechanism for the protective effect of Cyy-287 on HFD-induced inflammation.

Supplemental Information

Supplemental Information 1 Cyy-287 attenuates the mRNA expression of SREBP-1c in HFD mice.

Supplemental Information 2 Original data.

Supplemental Information 3 Checklist.

Additional Information and Declarations

Competing Interests

Author Contributions

Animal Ethics

Data Availability

The authors declare that they have no competing interests.

Jinhuan Ni performed the experiments, analyzed the data, prepared figures and/or tables, and approved the final draft.

Xiaodan Zhang performed the experiments, prepared figures and/or tables, and approved the final draft.

Huijing Huang performed the experiments, prepared figures and/or tables, and approved the final draft.

Zefeng Ni performed the experiments, prepared figures and/or tables, and approved the final draft.

Jianchao Luo analyzed the data, prepared figures and/or tables, and approved the final draft.

Yunshan Zhong analyzed the data, prepared figures and/or tables, and approved the final draft.

Min Hui analyzed the data, prepared figures and/or tables, and approved the final draft.

Zhiguo Liu conceived and designed the experiments, authored or reviewed drafts of the article, and approved the final draft.

Jianchang Qian conceived and designed the experiments, authored or reviewed drafts of the article, and approved the final draft.

Qianwen Zhang conceived and designed the experiments, authored or reviewed drafts of the article, and approved the final draft.

The following information was supplied relating to ethical approvals (i.e., approving body and any reference numbers):

The studies were approved on 20-May-2019 by the NIH Guide for the Care and Use of Laboratory Animals approved by the Laboratory Animal Ethics Committee of Wenzhou Medical University for which the Approval ID is WYDW-2019-0809.

The following information was supplied regarding data availability:

The raw measurements are available in the Supplemental Files.

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
