# Peer review of "Cyy-287, a novel pyrimidine-2,4-diamine derivative, efficiently mitigates inflammatory responses, fibrosis, and lipid synthesis in obesity-induced cardiac and hepatic dysfunction"

_PeerJ, doi:10.7717/peerj.17009_

## Round 0.1 · original submission · Major Revisions

Please address the concerns of all reviewers and amend the manuscript accordingly.

**Language Note:** PeerJ staff have identified that the English language needs to be improved. When you prepare your next revision, please either (i) have a colleague who is proficient in English and familiar with the subject matter review your manuscript, or (ii) contact a professional editing service to review your manuscript. PeerJ can provide language editing services - you can contact us at copyediting@peerj.com for pricing (be sure to provide your manuscript number and title). – PeerJ Staff

·

Basic reporting

Authors clearly explained NF-kB and AMPK signaling pathways and derived hypothesis based on their previous research. Their novel compound, Cyy-287, has a great potential for impeding obesity-induced deterioration of cardiac and hepatic functions. They demonstrated well signaling axes related to HFD induced inflammation with supporting data. Also, this compound is worthy to look into further for other disease such as type 2 diabetes.

Experimental design

1) At line 124,
a) the sum of fat, protein, and carbohydrate is 50%. What is the rest of 50% for the standard control diet?
b) Please briefly describe ‘adaptive feeding’ strategy

2) In their previous study (doi: 10.3724/abbs.2022139), authors administrated 15mg/kg of cyy-287 via i.p. and showed about less than 5% of the compound was found in the liver. In this study, the compound was administrated by gavage and the max dose is 20mg/kg. Given the previous study that shows low affinity for liver, why did author choose the administration method? Did author measured cyy-287 distribution in other organ? How did they come up with the max dose?

Validity of the findings

1) JNK has been reported to drive insulin resistance as well as metabolic inflammation in obesity (doi: 10.2147/DMSO.S236127). It would have greater potential if authors can show JNK expression level.

2) Please fix the typo though => through. ‘Cyy-287 mitigates hepatic fat accumulation and damage though p38 MAPK/NF- kB dowregulation and activating AMPK in HFD mice

Additional comments

no comment

Reviewer 2 ·

Basic reporting

This study demonstrates the protective effect and molecular mechanism of CYY-287 in mice fed with high fat diet. While this study is interesting and important, this can be improved by more complete characterization of mice and addressing a few key points. The following suggestions are provided to help strengthen this work.
1. Please include extra keywords like hepatic steatosis, etc.
2. Please clarify when the ECHO reading was recorded. Was it done at different timepoints or weeks?
3. It is important to include details on primary, secondary antibodies dilutions, catalog number, company info etc.
4. It is ideal to include an SREBP1-C gene panel expression analysis by RTPCR to understand the molecular mechanism better and its interrelation with DNL.
5. Were the TG, TC and lipid parameters measured in liver and heart?
6. The serum TG measurement is missing. It is ideal to include the analysis of FFA for clarity.
7. Please include liver weights, gross images of livers.
8. It is ideal to include more ECHO parameters. For example, LV function, output etc.
9. Figure 3D collagen 1 and P-p65 is not quite clear. It is ideal to improve the resolution for better clarity.

Experimental design

.

Validity of the findings

.

Reviewer 3 ·

Basic reporting

This study investigates the protective effects of a novel pyrimidine-2,4-diamine derivative, Cyy-287, in high-fat diet (HFD) mice experiencing inflammation and metabolic disorders associated with obesity. Results showed that Cyy-287 administration improved heart function, reduced serum glucose and lipid markers, and inhibited liver and heart histopathological changes. The study suggests that Cyy-287 could be a potential therapeutic drug for mitigating obesity-induced heart and liver damage by targeting inflammation and lipid synthesis pathways.

Experimental design

Line 122-123, HFD consisted of 60% fat, 20% protein, and 20% carbohydrate, while control mice were fed 10% fat, 20% protein and 20% carbohydrate. For the control mice, what is the other 50% of their diet?

Line 173, CYP450 should be assigned with a full name when first occurred.

Validity of the findings

The study mentions different dosages of Cyy-287, but it doesn't explain why these specific doses were chosen. The lack of a clear justification for the selected doses raises questions about the robustness of the reported results.

While statistical significance is mentioned in various parts of the results, the actual statistical methods used and the significance levels are not thoroughly explained. Please add the detailed statistical methods used.

---

## Round 0.2 · accepted · Accept

All concerns of the reviewer were adequately addressed and the manuscript was revised accordingly. The amended version is acceptable now.

Reviewer 3 ·

Basic reporting

Thanks for the author's responses to the comments, and the revision looks good to me.

Experimental design

N/A

Validity of the findings

N/A

Additional comments

N/A